# Five decades (1972–2020) of zooplankton monitoring in the upper San Francisco Estuary

**Samuel M. Bashevkin**[1]*, **Rosemary Hartman**[2], **Madison Thomas**[1], **Arthur Barros**[3], **Christina E. Burdi**[3], **April Hennessy**[4], **Trishelle Tempel**[3], **Karen Kayfetz**[1]

**1** Delta Science Program, Delta Stewardship Council, Sacramento, California, United States of America, **2** California Department of Water Resources, West Sacramento, California, United States of America, **3** California Department of Fish and Wildlife, Stockton, California, United States of America, **4** California Department of Fish and Wildlife, West Sacramento, California, United States of America

* sam.bashevkin@deltacouncil.ca.gov

## Abstract

We present the longest available dataset (by 15 years) of estuarine zooplankton abundance worldwide. Zooplankton have been monitored throughout the upper San Francisco Estuary from 1972 –present due to its status as a central hub of California water delivery and home to commercially important and endangered fishes. We integrated data from five monitoring programs, including over 300 locations, three size-classes of zooplankton targeted with different gears, over 80,000 samples, and over two billion sampled organisms. Over the duration of this dataset, species invasions have driven community turnover, periodic droughts have occurred, and important fishes have declined, likely due in part to reduced food supply from zooplankton. Data from the individual surveys have been used in prior studies on issues related to species invasions, flows, fish diets and population dynamics, zooplankton population dynamics, and community ecology. Our integrated dataset offers unparalleled spatio-temporal scope to address these and other fundamental ecological questions.

## Introduction

### Background and motivation

Zooplankton are a critical component of aquatic communities, comprising a crucial link between primary production and upper trophic taxa of high management interest such as fishes. Zooplankton are also indicator species for climate change [1], can exert top-down control on primary production [2], and are often caught and enumerated in high quantities making them ideal candidates for synthetic analyses.

Long-term monitoring data are often the best datasets to answer pressing management questions and build ecological theory [3]. These datasets offer the opportunity to thoroughly test hypotheses and understand ecological processes over the high variability of biological and environmental data [4]. Long-term datasets are necessary to evaluate changes resulting from long-term patterns such as climate change and climate cycles, or to evaluate the effects of management actions or disturbances against a background of high inter-annual variability [4]. Lastly, these datasets are necessary to train predictive models for ecological forecasting [5].

**Data Availability Statement:** All data files are available on the Environmental Data Initiative (doi:10.6073/pasta/89dbadd9d9dbdfc804b160c81633db0d and web

link: https://portal.edirepository.org/nis/mapbrowse?scope=edi&identifier=539&revision=3).

**Funding:** The author(s) received no specific funding for this work.

**Competing interests:** The authors have declared that no competing interests exist.

Long-term monitoring of zooplankton communities are common in lake systems, where long-term zooplankton monitoring has been conducted in Germany since 1979 [6], Estonia since 1950 [7], and Japan since 1980 [8]. There are also several long-term marine zooplankton data sets, such as the Continuous Plankton Recorder Survey in the North Atlantic which started in 1931 [9, 10], and the California Cooperative Oceanic Fisheries Investigations which started in 1951 [11].

However, long-term zooplankton datasets from estuaries are less common, and mostly of shorter time scales or smaller spatial scope. Zooplankton have been monitored long-term in Narragansett Bay, Rhode Island, USA since 1972 (https://web.uri.edu/gso/research/plankton), the Gironde Estuary, France since 1978 [12, 13], the North Inlet Estuary, South Carolina, USA since 1981 [14], and the Hudson River, New York, USA since 1987 [15]. However, each of these programs only sample at one or two locations and, with the exception of the Hudson River, their data are either unavailable or only available for a few years of the time series (e.g., [16]). The Chesapeake Bay Program, USA started monitoring zooplankton in 1982 but this program was discontinued in 2002 [17]. Similarly, the Bristol Channel Project (United Kingdom) monitored plankton from 1971–1983 (https://www.bodc.ac.uk/resources/products/data/bodc_products/bristol/). Other estuaries began monitoring zooplankton more recently, such as the Boston Harbor, Massachusetts, USA since 1992 [18]; Kongsfjorden, Norway since 1996 [19]; the Scheldt estuary, Belgium since 1996 [20]; the Mondego estuary, Portugal since 2003 [21]; the St. Lawrence Estuary, Canada since 2010 [22]; and the Puget Sound, Washington, USA since 2014 (https://green2.kingcounty.gov/marine/Monitoring/Zooplankton). The Pacific Northwest estuaries program has monitored zooplankton from a number of estuaries in the USA and Canada since 1996, but sampling is only conducted every four years [23].

Long-term zooplankton monitoring is especially important in estuaries given their economic and ecological importance. Estuaries provide productive nursery grounds for fisheries, unique species assemblages, and are key areas of human development and trade [24–26]. However, estuaries are highly variable systems over spatial and temporal scales as the salinity field shifts with tides or longer-term seasonal to decadal drivers [27]. These natural changes, as well as rapid changes associated with human use of estuaries, such as changes to freshwater flow, sediment supply, nutrient loads, contaminants, climate change, and invasive species mean estuaries are much more dynamic than many other aquatic systems [28]. Long-term monitoring with a spatially distributed sampling design is critical to capturing and accounting for the full range of environmental drivers.

The upper San Francisco Estuary (SFE), USA is an estuary of high economic and ecological importance. The upper SFE is comprised of the Sacramento-San Joaquin Delta (Delta), as well as San Pablo Bay (the northern portion of San Francisco Bay), Suisun Bay, and Suisun Marsh (Fig 1). The Delta is an inland inverse delta formed by the confluence of five major rivers that drain 40% of the land in California [29]. Two water projects, state (State Water Project) and federal (Central Valley Water Project), supply fresh water from the Delta to two-thirds of the state's population and to irrigate millions of acres of farmland [29]. Agricultural lands and cities surrounding and within the Delta also divert and input water (and contaminants) into the aquatic system. The upper SFE is home to several commercially and recreationally important fisheries and five fish species listed under the United States Endangered Species Act and/or the California Endangered Species Act: Chinook Salmon (*Oncorhynchus tshawytscha*), Green Sturgeon (*Acipenser medirostris*), Delta Smelt (*Hypomesus transpacificus*), Longfin Smelt (*Spirinchus thaleichthys*), and Steelhead Trout (*Oncorhynchus mykiss*). These fishes utilize the full salinity range of the upper SFE, from the fresh Delta to the saltwater San Pablo Bay, and most rely on zooplankton prey in their juvenile life stages. Delta Smelt are endemic to the upper SFE and depend on zooplankton for prey throughout their lifecycle [30]. To monitor the

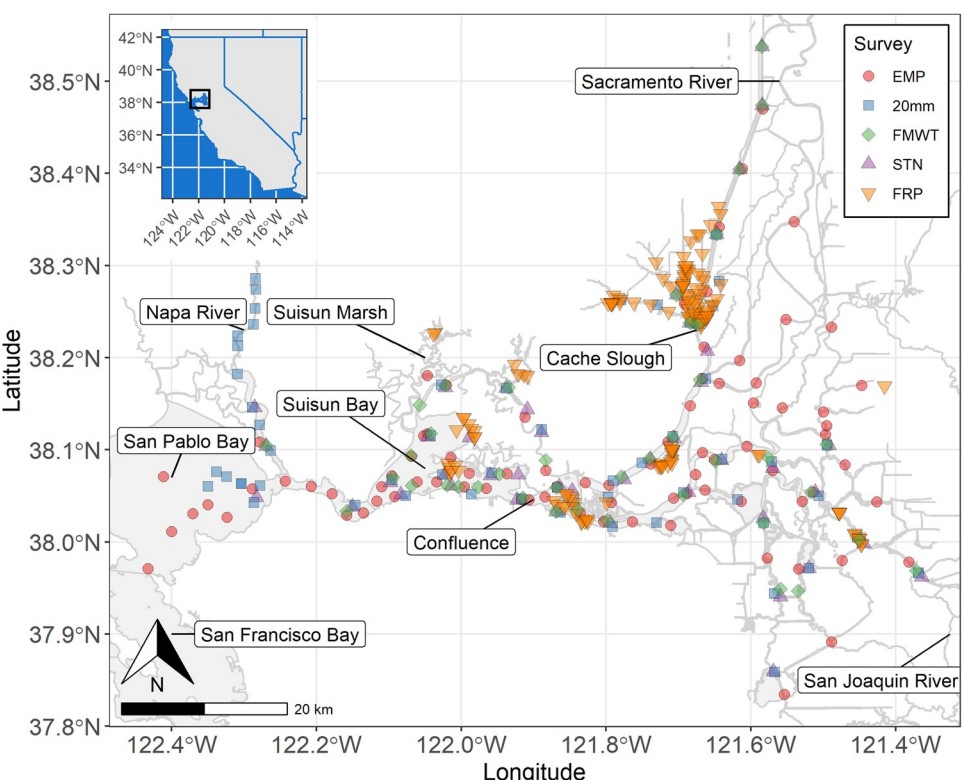

**Fig 1. Map of active and historical sampling locations.** Only fixed stations are shown. The unfixed EMP EZ station locations move with the estuarine salinity gradient and are provided in the stations_EMP_EZ table. The Sacramento San Joaquin Delta is the region East of the Confluence. State boundaries were reproduced from the United States Census Bureau.

environmental impacts of water exports from this system, extensive ecological monitoring has been conducted since the 1960s and zooplankton abundance has been monitored since 1972, resulting in a zooplankton sampling program of unparalleled temporal and spatial scale. Zooplankton have been sampled since 1972 at numerous sampling stations that have expanded over the years, covering a 3,500 km$^2$ region. There are 311 total fixed sampling stations, 12 of which have data for 46 years or more, and 89 with at least 20 years of data.

The upper SFE has undergone extensive changes over the duration of these monitoring programs, offering abundant opportunities to address globally relevant questions with this dataset. Numerous species have invaded, making the SFE the most invaded estuary in the world [31], and transforming the zooplankton community into one mostly East Asian in origin [32]. These invasions were facilitated by climatic extremes and hydrological management, providing a case study for the intersection between climate changes and environmental management [33]. Water exports have steadily increased [34] while inflow from upstream watersheds have been highly variable year-to-year, and that variability is projected to increase with climate change [35]. Then, in 2002 a number of fish species dramatically declined during what has been called the "Pelagic Organism Decline" [36, 37], likely due in part to reduced zooplankton food supply [30, 38]. This short-term crash of important fish species was part of a larger, long-term pattern of environmental decline in human-impacted estuaries across the world [39]. Lastly, the upper SFE represents one of the largest historical losses of wetland habitat, with 97% of wetlands lost since the early 1800s [40]. More recently, numerous restoration efforts have been slowly increasing wetland habitat in the region, mostly aimed at increasing

habitat and food for fishes [30]. Long-term wetland decline is a global phenomenon, and over half of global wetland area has been lost since 1700 [41].

The unparalleled temporal scale and spatial distribution of our integrated dataset will provide great fodder for ecological synthesis. With the range of natural variability and past events captured in this dataset, it can be leveraged to answer broad questions in invasion ecology, aquatic and marine ecology, climate change ecology, and population biology. We hope it will also spur research to inform management of the SFE. The upper SFE has a great body of zooplankton research, including many studies using data from the component surveys included in our integrated dataset. However, key questions of management importance remain unanswered, particularly in relation to drivers of fish food abundance and quality [42]. Our integration of multiple monitoring surveys will make this dataset easier to use to inform management in the SFE, while the publication and description of this dataset to a broader audience will provide an analysis-ready dataset of estuarine zooplankton for ecological synthesis.

One issue with the analysis of long-term plankton monitoring data is that inconsistencies in taxonomic resolution can make it difficult or impossible to conduct unbiased analyses [43]. Fortunately, all five surveys included in this integrated dataset had documented the taxonomic resolution at which samples were counted in each year. While changes to taxonomic methods persisted in the underlying datasets, we developed an approach to correct these issues in the final integrated dataset and produce an analysis-ready dataset with consistent taxonomic resolution. The methods we have developed for accounting for changes in taxonomic resolution between surveys and over time are applicable to any zooplankton data set, allowing other researchers to adopt similar methods for datasets from other regions, and paving the way for broader analyses of zooplankton across habitats and around the world.

## Monitoring overview

This dataset is a compilation of zooplankton abundance and associated environmental data from five California Department of Fish and Wildlife (CDFW) monitoring surveys conducted under the auspices of the Interagency Ecological Program. The Interagency Ecological Program (IEP) is a consortium of state and federal agencies that has been conducting cooperative ecological investigations in the SFE since the 1960s. The IEP runs more than twenty long-term monitoring surveys on biological components of the upper SFE. Surveys monitor phytoplankton, zooplankton, benthic invertebrates, water quality, and fishes.

Starting with the Environmental Monitoring Program (EMP) [44, 45] run collaboratively by CDFW, the California Department of Water Resources, and the United States Bureau of Reclamation, the suite of zooplankton surveys gradually expanded with time. The CDFW 20mm survey [46] started in 1995 to survey postlarval and juvenile Delta Smelt and included zooplankton sampling. In the 2000s, two CDFW surveys traditionally focused on monitoring fish abundance added zooplankton nets to their sampling programs to investigate potential linkages between food supply and declining fish abundance. The Summer Townet Survey (STN) [47] added zooplankton sampling in 2005 and the Fall Midwater Trawl (FMWT) [47] added zooplankton sampling in 2011, with pilot studies conducted in earlier years (back to 2005). Lastly, a new monitoring program, the Fish Restoration Program (FRP) [48], began in 2016, sampling both fish and zooplankton (Fig 2, Table 1). All surveys are ongoing.

Each survey samples monthly or once every two weeks at sets of fixed stations that vary among surveys depending on their objectives. While EMP samples year-round, the other surveys are mostly seasonal (Fig 2, Table 1). However, sampling effort has fluctuated historically due to changing management needs, unforeseen events, or shifting priorities. For example, EMP initially sampled every two weeks before shifting to monthly sampling in the mid-1990s

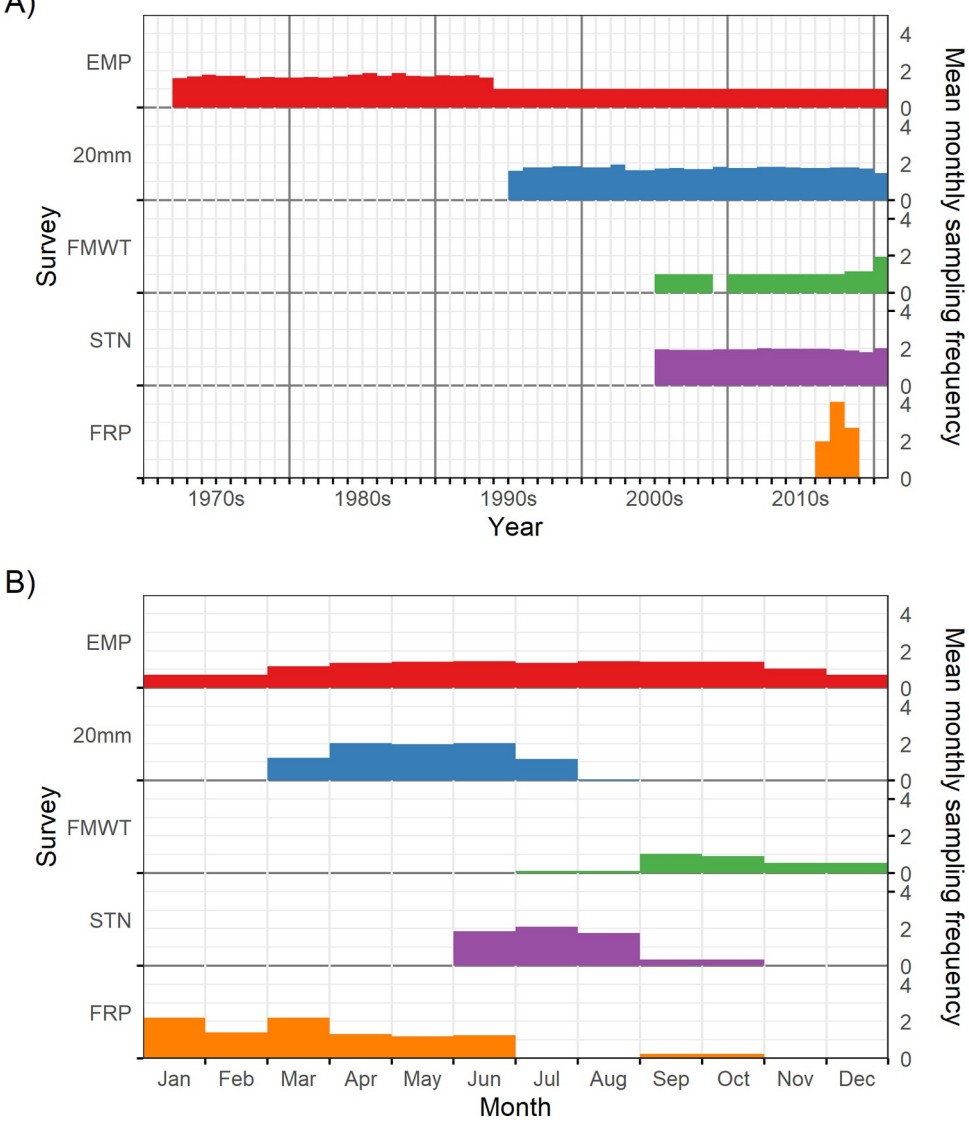

**Fig 2. Seasonal and yearly sampling effort for each survey.** Sampling effort was quantified as the mean number of samples collected per station for each month of each year, then averaged at A) yearly and B) monthly timesteps. All surveys continue today, so the final year reflects the latest date in the released data, rather than the end of sampling. FRP had high and variable sampling effort due to experimental sampling to refine techniques in the first years of this new survey.

(Fig 2). Occasionally, special studies have added sampling effort to these long-term programs. One example is the Suisun Marsh Salinity Control Gate Study, a special study to monitor the effects of a management action (the Suisun Marsh Salinity Control Gate action) on food resources from July to October 2018 [49]. This effort temporarily increased spatiotemporal sampling effort for the FMWT and STN and added macrozooplankton sampling to the STN samples included in the special study. The Suisun Marsh Salinity Control Gate Study data are included in the integrated dataset as FMWT or STN data depending on the identity of the sampled station.

Most surveys target open channels although FRP samples in or near shallow tidal wetlands. Three size classes of zooplankton are targeted by these sampling programs with different net

**Table 1. Sampling design and start years for environmental variables measured by each monitoring survey.**

|  | EMP | 20mm | FMWT | STN | FRP |
|---|---|---|---|---|---|
| First year | 1972 | 1995 | 2005 | 2005 | 2016 |
| Frequency | Monthly | 2 weeks | Monthly | 2 weeks | Monthly |
| Time of year | All year | Mar-Jul | Sep-Dec | Jun-Aug | Mar-Dec |
| Tidal stage | High slack | Variable | Variable | Variable | Variable |
| Conductivity (S) | 1972 | 1995 | 2005 | 2005 | 2016 |
| Conductivity (B) | 1981 | 1995 | 2006 | 2006 |  |
| Temperature | 1972 | 1995 | 2005 | 2005 | 2016 |
| Secchi depth | 1972 | 1995 | 2005 | 2005 | 2016 |
| Turbidity |  | 2011 | 2010 | 2010 | 2016 |
| *Microcystis* sp. |  |  | 2007 | 2007 | 2016 |
| Chlorophyll | 1976 |  |  |  |  |
| pH |  |  |  |  | 2016 |
| Dissolved oxygen |  |  |  |  | 2016 |

The start years are indicated for each environmental variable, while missing values indicate variables not sampled by that survey. (S) and (B) represent surface and bottom samples of conductivity, respectively. A frequency of "2 weeks" indicates that sampling is conducted once every two weeks. More information is available in study_metadata.csv of the published dataset. EMP refers to the Environmental Monitoring Program, 20mm refers to the 20-mm survey, FMWT refers to the Fall Midwater Trawl, STN refers to the Summer Townet, and FRP refers to the Fish Restoration Program.

mesh sizes: microzooplankton (copepods and rotifers) are targeted with a pump discharging into a 43 μm mesh net, mesozooplankton (copepods and cladocerans) are targeted with 150–160 μm mesh nets, and macrozooplankton (mysids and amphipods) are targeted with 500–505 μm mesh nets. All surveys use at least a mesozooplankton net, which targets adult copepods and cladocerans, because these taxa are believed to comprise the majority of zooplankton in planktivorous fish diets [50–54] (Table 2).

Along with zooplankton counts, our integrated dataset also includes environmental data. Data from all surveys include surface conductivity, temperature, and Secchi depth since their inception. Some surveys also include bottom conductivity, turbidity, pH, chlorophyll-a, dissolved oxygen, and concentration of the harmful alga *Microcystis* sp. for some years (Table 1).

**Table 2. Current sampling methods for each gear type and survey.**

| Survey | Size class | Mesh size | Method | Duration | First year |
|---|---|---|---|---|---|
| EMP | Macro | 505 μm | oblique tow | 10 min | 1972 |
| EMP | Meso | 160 μm | oblique tow | 10 min | 1972 |
| EMP | Micro | 43 μm | vertical pump | 0.075 m$^3$ | 1972 |
| 20mm | Meso | 160 μm | oblique tow | 10 min | 1995 |
| FMWT | Macro | 505 μm | oblique tow | 10 min | 2007 |
| FMWT | Meso | 160 μm | oblique tow | 10 min | 2005 |
| STN | Meso | 160 μm | oblique tow | 10 min | 2005 |
| FRP | Macro | 500 μm | horizontal tow | 5 min | 2016 |
| FRP | Meso | 150 μm | horizontal tow | 5 min | 2016 |

Sampling duration is measured in time (min) for net tows and volume (m$^3$) for pump samples (EMP micro). Some methods have changed over time, see *Methods* for descriptions of these changes. More information on each survey is available in study_metadata.csv of the published dataset. The first year indicates the first year data were collected, but may be a pilot year with more regular sampling starting later. See Fig 2 for more information on the annual sampling effort of each survey. EMP refers to the Environmental Monitoring Program, 20mm refers to the 20-mm survey, FMWT refers to the Fall Midwater Trawl, STN refers to the Summer Townet, and FRP refers to the Fish Restoration Program.

While all led or co-led by CDFW, each monitoring program is independently managed. Most methods are similar, but key differences remain (Tables 1 and 2). Most importantly, the list of taxa identified in samples slightly differs in each study. To resolve these issues and improve the usability of data from all five surveys, we documented methodological differences among the surveys, integrated their data, and developed a standardized method for resolving differences in taxonomic resolution. This integrated zooplankton dataset offers much greater spatiotemporal resolution than can be found in any single survey. The spatial extent of the integrated dataset includes the Delta, Suisun Bay and Marsh, Napa River, and San Pablo Bay (Fig 1). This encompasses the full salinity range of the upper SFE, from saltwater in San Pablo Bay to freshwater in the Sacramento and San Joaquin Rivers.

## Dataset description and access

The integrated dataset [55] includes a series of tables connected by keys that include sampling station locations (stations.csv and stations_EMP_EZ.csv), sample-level environmental and datetime data (environment.csv), information on taxa poorly sampled by micro and meso-zooplankton nets (undersampled.csv), biomass conversion values (biomass_mesomicro.csv), taxonomic information (taxonomy.csv) and zooplankton abundance data as catch per unit effort (zooplankton.csv). These tables comprise the raw integrated dataset. Also included are a pre-joined dataset including all variables from the prior tables and with taxonomic issues resolved (see methods; zooplankton_community.csv), a table listing the taxa counted by each survey (taxa_lists.csv), and a table with detailed metadata information on each component survey (study_metadata.csv; Fig 3, Tables 3 and 4).

In addition to the data publication [55], the integrated dataset may also be accessed via the R package zooper [56], which offers more options for customizing the data integration and taxonomic resolutions. An associated R shiny application is available at https://DeltaScience. shinyapps.io/ZoopSynth. The shiny application offers an interactive point-and-click interface

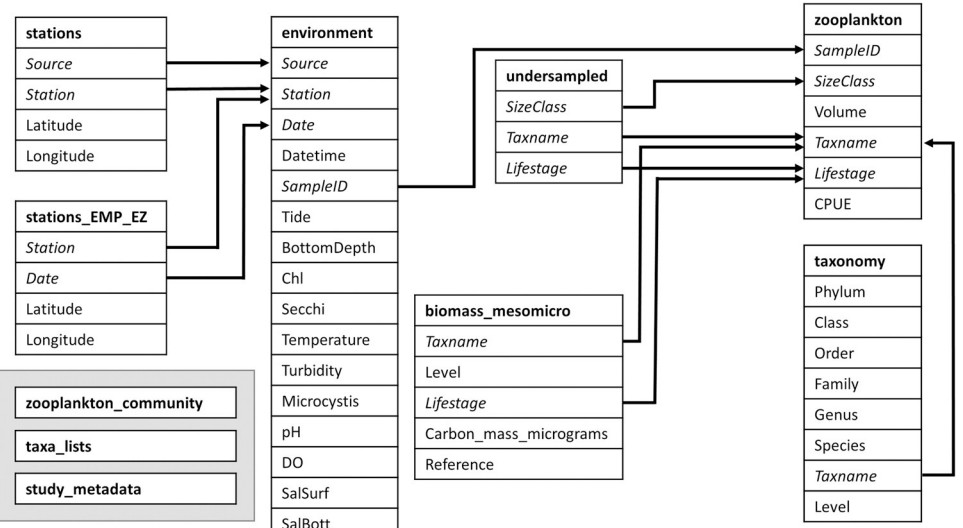

**Fig 3. Linkages among the main data tables.** Table names are in bold and keys are in italics and connected by arrows. Only tables linked to the zooplankton table are shown in full detail, others are listed in the gray box. The zooplankton_community table is composed of the other linked tables (except biomass_mesomicro) joined together and its taxonomic resolution has been standardized. The taxa_lists table contains the list of taxa identified in each survey and net size, along with dates when taxa lists changed and species introduction years. The study_metadata table contains information on each component survey. Descriptions of variables are given in Tables 3 and 4.

**Table 3. Descriptions of variables in the zooplankton data tables.**

| Variable | Description |
| --- | --- |
| Source | Name of the source dataset |
| Station | Station where sample was collected |
| Latitude | Latitude of station location (decimal degrees) |
| Longitude | Longitude of station location (decimal degrees) |
| Year | Year sample was collected (YYYY) |
| Date | Date sample was collected (YYYY-MM-DD) |
| Datetime | Date and time sample was collected (YYYY-MM-DD hh:mm:ss) |
| SampleID | Unique ID of each zooplankton sample |
| Tide | Tidal stage |
| BottomDepth | Total water column depth (m) |
| Chl | Chlorophyll concentration at surface ($\mu g\ L^{-1}$) |
| Secchi | Secchi depth (cm) |
| Temperature | Surface temperature (˚C) |
| Turbidity | Turbidity at surface (NTU) |
| Microcystis | *Microcystis* sp. abundance ranking on scale from 1–5 (absent, low, medium, high, very high) |
| pH | Surface pH |
| DO | Surface dissolved oxygen concentration ($mg\ L^{-1}$) |
| SalSurf | Surface salinity |
| SalBott | Bottom salinity |
| SizeClass | Zooplankton size class |
| Volume | Volume of water sampled ($m^3$) |
| Phylum | Phylum |
| Class | Class |
| Order | Order |
| Family | Family |
| Genus | Genus |
| Species | Species |
| Level | Taxonomic level of taxon |
| Taxname | Scientific name of taxon |
| Lifestage | Life stage |
| Taxlifestage | Scientific name and life stage |
| CPUE | Catch per unit effort |
| Undersampled | This count represents an under-sample of the true concentration of this taxa in the water sampled |
| Carbon_mass_micrograms | Average carbon mass of an individual ($\mu g$) |
| Reference | Source of biomass value |
| EMP_Micro | Taxon counted in the EMP microzooplankton samples |
| EMP_Meso | Taxon counted in the EMP mesozooplankton samples |
| EMP_Macro | Taxon counted in the EMP macrozooplankton samples |
| STN_Meso | Taxon counted in the STN mesozooplankton samples |
| STN_Macro | Taxon counted in the STN macrozooplankton samples |
| FMWT_Meso | Taxon counted in the FMWT mesozooplankton samples |
| FMWT_Macro | Taxon counted in the FMWT macrozooplankton samples |
| twentymm_Meso | Taxon counted in the 20mm mesozooplankton samples |
| FRP_Meso | Taxon counted in the FRP mesozooplankton samples |
| FRP_Macro | Taxon counted in the FRP macrozooplankton samples |
| Intro | If nonnative, year species was introduced to the San Francisco Estuary |
| EMPstart | First year this taxon was counted in EMP samples (YYYY) |

*(Continued)*

**Table 3.** (Continued)

| Variable | Description |
|---|---|
| EMPend | Last year this taxon was counted in EMP samples (YYYY) |
| FMWTSTNstart | First year this taxon was counted in FMWT and STN samples (YYYY) |
| FMWTSTNend | Last year this taxon was counted in FMWT and STN samples (YYYY) |
| twentymmstart | First year this taxon was counted in 20mm samples (YYYY) |
| twentymmend | Last year this taxon was counted in 20mm samples (YYYY) |
| twentymmstart2 | Year this taxon was re-added to the taxa list and counted in 20mm samples (YYYY) |

These variable descriptions apply to all data tables except study_metadata.csv, which is described in Table 4. EMP refers to the Environmental Monitoring Program, 20mm refers to the 20-mm survey, FMWT refers to the Fall Midwater Trawl, STN refers to the Summer Townet, and FRP refers to the Fish Restoration Program.

to zooper, as well as data visualizations, and should be useful for both non-R-programmers and programmers interested in quick data visualization and exploration.

## Methods

### Zooplankton surveys

**Environmental monitoring program.** The Environmental Monitoring Program (EMP) Zooplankton Study (also known as the IEP Zooplankton Study) began in 1972 in order to assess trends in fish food resources ranging from San Pablo Bay to the east Delta, as well as to detect and assess the impacts of recently introduced zooplankton species on native species. The study is mandated by Water Right Decision 1641 for operation of the State Water Project and Central Valley Project [57]. The EMP study is conducted by the California Department of Fish and Wildlife (CDFW), the California Department of Water Resources, and the United States Bureau of Reclamation, and currently samples 17 fixed stations and 2–4 floating entrapment zone stations. Entrapment zone stations are locations where the bottom conductivity is 2 and 6 mS/cm. When these points occur upstream of the confluence of the Sacramento and San Joaquin rivers, two stations at each salinity point are taken, one on each river. There are also 3 additional stations located in Carquinez Strait and San Pablo Bay, which are sampled during periods of high outflow and low salinity. Historically (prior to 1995) the survey sampled more frequently and at many more stations (Fig 2).

EMP samples zooplankton in all three size ranges: microzooplankton, mesozooplankton, and macrozooplankton. Zooplankton are collected monthly at fixed stations year-round in open channels at high slack tide and preserved in 10% formalin dyed with rose bengal. Macrozooplankton and mesozooplankton are collected using a mysid net and a Clarke-Bumpus net, respectively, during 10-minute oblique tows. The mysid net is 124 cm long with a 28 cm diameter mouth and 505 μm mesh, while the Clark-Bumpus net is 73 cm long with a 12 cm diameter mouth and 160 μm mesh. Both nets have a flowmeter mounted in the mouth and cod-ends with the same mesh size as the net. Prior to 1974, macrozooplankton were sampled with a 930 μm mesh net. Microzooplankton are collected with a Teel marine pump while the intake hose is raised through the water column and pumped into a net with 43 μm mesh. Pump samples collected approximately 1.5–1.9 L from 1972–2007, and 75 L from 2008—present, measured by a digital flowmeter connected to the hose.

Microzooplankton samples are passed through a 154 μm mesh sieve nested on top of a 43 μm mesh sieve in the laboratory, and only the smaller size fraction that passes through the larger sieve and is retained on the smaller sieve is counted. Lengths are recorded for macrozooplankton (but not yet included in this integrated dataset). Recorded environmental variables

**Table 4. Descriptions of variables in the study_metadata.csv table.**

| Variable | Description |
| --- | --- |
| Study_name | Name of the survey |
| Size_class | Zooplankton size class sampled |
| Contact_person | PI or contact person for the survey or study |
| Contact_email | Email for contact person |
| Link_to_data | Link to the data, if online, or other way to get the data |
| Link_to_info_on_study | Link to project website, if available |
| Start_year | Year program started |
| Frequency | Sample collection frequency |
| Time_of_year | Months in which sampling occur |
| San_Pablo_Bay | San Pablo Bay is sampled |
| Suisun | Suisun Bay is sampled |
| Sacramento_River | The Sacramento River is sampled |
| San_Joaquin_River | The San Joaquin River is sampled |
| Cache_Slough_Complex | The Cache Slough Complex is sampled |
| Napa_River | The Napa River is sampled |
| Tidal_stage_sampled | When on the tidal stage sampling occurs |
| Sampling_scheme | How stations are sampled (at fixed location or haphazardly near the fixed location) |
| Gear_type | Sample collection gear (net or pump) |
| Sample_duration_minutes | Net tow duration (minutes) |
| Sampling_method | Samples collection method (oblique, vertical, or horizontal) |
| Length_of_net_cm | Net length (cm) |
| Mesh_size_microns | Mesh size used (μm) |
| Habitat_sampled | Habitat where samples are collected (channels, shoals, shallow water, deep water, wetlands, etc) |
| Copepods | Copepods are counted |
| Rotifers | Rotifers are counted |
| Cladocera | Cladocerans are counted |
| Mysids | Mysids are counted |
| Amphipods | Amphipods are counted |
| Other_taxa | Any other taxa that are counted |
| Subsampling_method | How samples are divided for counting and the parameters used to decide how much of the sample to count. |
| Magnification | Microscope magnification during sample processing |
| Preservative | Sample preservation medium |
| CPUE_calculation | CPUE calculation formula |
| Biomass | Biomass can be estimated |
| Lengths_measured | Lengths are measured |
| Sample_archived | Samples are archived |
| Time | Time of day is recorded |
| Tidal_stage | Tidal stage is recorded |
| Depth_of_water | The total depth of the water is recorded |
| Surface_conductivity | Conductivity at the surface is recorded |
| Bottom_conductivity | Conductivity at the surface is recorded |
| Temperature | Water temperature is recorded |
| Secchi | Secchi depth is recorded |
| Turbidity | Water turbidity is recorded |
| Microcystis | *Microcystis* sp. abundance is recorded |
| Chlorophyll | Chlorophyll-a concentration is recorded |

(*Continued*)

**Table 4.** (Continued)

| Variable | Description |
|---|---|
| pH | Water pH is recorded |
| DO | Dissolved oxygen concentration is recorded |
| Volume | The total volume of water filtered through the net is recorded |

include time, depth, surface and bottom specific conductivity, surface temperature, Secchi depth, and chlorophyll-a (Table 1).

More information on EMP and its methods can be found on the EMP zooplankton study website, zooplankton data publication [44], or environmental data publication [45].

**20mm survey.** The 20-mm Survey was initiated in 1995 by the California Department of Fish and Wildlife to monitor postlarval-juvenile Delta Smelt (*Hypomesus transpacificus*) distribution, abundance, and timing throughout their historical spring range in the Delta. The survey supports compliance of the Endangered Species Act Biological Opinion for operation of the State and Central Valley water projects [58]. 20-mm refers to the length of the fish targeted by the net. Zooplankton samples are collected concurrently with fish samples to monitor Delta Smelt food supply. Between 41 and 55 stations have been sampled each year since the survey began. Stations are spread across the Delta, Suisun Bay, eastern San Pablo Bay, and the Napa River (Fig 1).

Zooplankton are sampled every two weeks between March and July at fixed stations in open channels (Fig 2). Mesozooplankton are sampled using 10-minute stepped-oblique tows with a 73 cm long 160 μm mesh modified Clarke-Bumpus net. The net is attached to the top of the 20-mm Survey net frame and a flowmeter is mounted in the mouth. Samples are preserved in 10% formalin. Recorded environmental variables include times, tidal stage, depth, surface and bottom conductivity, surface temperature, Secchi depth, and turbidity (Table 1).

More information on the 20-mm Survey and its methods can be found on the 20-mm Survey website [46].

**Fall Midwater Trawl.** The Fall Midwater Trawl (FMWT) was initiated by the California Department of Fish and Wildlife in 1967 in order to determine the relative abundance and distribution of age-0 Striped Bass (*Morone saxatilis*), but the data have also been used for other upper estuary pelagic fish species, including Delta Smelt, Longfin Smelt, American Shad (*Alosa sapidissima*), Splittail (*Pogonichthys macrolepidotus*), and Threadfin Shad (*Dorosoma petenens*e). The FMWT supports compliance of the 2019 Delta Smelt Biological Opinion for the coordinated operation of the Central Valley Project and State Water Project [58]. The FMWT samples 122 stations each month from September to December ranging from San Pablo Bay to Stockton, Hood, and the Sacramento Deep Water Ship Channel (Figs 1 and 2). FMWT samples both macrozooplankton and mesozooplankton at a subset of these stations since 2011, with some pilot studies in earlier years back to 2005.

Zooplankton samples are collected immediately before or after the fish trawl at fixed stations in open channels using 10-minute stepped-oblique tows. Macrozooplankton are sampled using a 124 cm long net with 505 μm mesh, while mesozooplankton are sampled using a 73 cm long modified Clark-Bumpus net with 160 μm mesh. For both zooplankton sizes, samples are preserved in 10% formalin dyed with rose bengal. Lengths are recorded for macrozooplankton as in EMP (but not yet included in this integrated dataset). Recorded environmental variables include time, tidal stage, depth, surface and bottom conductivity, surface temperature, Secchi depth, presence of the harmful alga *Microcystis* sp., and turbidity (Table 1).

More information on FMWT and its methods can be found on the FMWT website [47].

**Summer Townet Survey.** The Summer Townet Survey (STN) was initiated by the California Department of Fish and Wildlife in 1959 in order to determine the relative abundance and distribution of upper estuary pelagic species, namely age-0 Striped Bass (*Morone saxatilis*). As with the FMWT and 20-mm Survey, the STN supports compliance of the 2019 Delta Smelt Biological Opinion [58] and began in response to the development of the Central Valley Project pumping plants. The STN collects mesozooplankton samples from 32 historic stations and eight supplemental stations ranging from San Pablo Bay to Rio Vista, Stockton, Cache Slough, and the Sacramento Deep Water Ship Channel (Fig 1). Zooplankton monitoring began in 2005 with samples collected every two weeks between June and August (Fig 2).

STN samples only mesozooplankton during their fish trawl with a net attached to the townet frame (although supplemental macrozooplankton sampling using FMWT methods was temporarily added in 2018 as part of the Suisun Marsh Salinity Control Gate Study [49]). Zooplankton samples are collected during one of the fish tows at each fixed station in open channels using 10-minute oblique tows. Mesozooplankton are sampled using a 73 cm long modified Clark-Bumpus net with 160 μm mesh and preserved in 10% formalin dyed with rose bengal. Recorded environmental variables include time, tidal stage, depth, surface and bottom conductivity, surface temperature, Secchi depth, presence of the harmful alga *Microcystis* sp., and turbidity (Table 1).

More information on STN and its methods can be found on the STN website [47].

**Fish Restoration Program.** The Fish Restoration Program (FRP) is devoted to restoring 8,000 acres of tidal habitat in the Delta and Suisun Marsh to provide Delta Smelt habitat and 800 acres of low salinity habitat to benefit Longfin Smelt. These restoration projects are pursuant to requirements in the 2019 Biological Opinions for State and Central Valley water project operations [58]. The FRP Monitoring Team monitors fish and their food resources (including zooplankton) within these restored wetlands in order to better understand the benefits of the restored habitats to native fish species. The FRP Monitoring Team surveys zooplankton in shallow waters, generally near tidal marshes or sites that will soon be converted to tidal marsh. FRP has worked closely with some other IEP surveys to compare zooplankton communities in shallow water to the open-water channel samples collected by the long-term surveys [59].

Zooplankton have been sampled annually to monthly between March and December beginning in 2016 (Fig 2). Samples are taken from haphazardly selected locations within fixed sites at restored and existing wetlands and adjacent open-water areas across the Delta and Suisun Marsh (Fig 2). Macrozooplankton are collected with 10-minute horizontal surface tows using a 0.4 m x 0.4 m mouth net (500 μm mesh size). Mesozooplankton are collected with 5-minute surface tows using a 14.6 cm diameter net (150 μm mesh size). Flowmeters are attached to both nets. Samples are preserved in 70% ethanol with rose bengal. Lengths are recorded for macrozooplankton (but not yet included in this integrated dataset). Recorded environmental variables include time, tidal stage, surface conductivity, surface temperature, Secchi depth, turbidity, presence of the harmful alga *Microcystis* sp., pH, chlorophyll, and dissolved oxygen (Table 1).

More information on FRP and its methods can be found on the FRP data publication [48].

## Descriptions of common methods

**Mesh sizes.** Nets/sieves typically sample zooplankton whose smallest dimension is larger than the mesh size, but may also capture some organisms smaller than the mesh size (which are under-sampled since some of these smaller plankters are washed through the mesh). Furthermore, organisms significantly larger than the net mesh may be able to avoid the net and thereby evade capture. Lastly, sample processing methods like the EMP sieving of microzooplankton samples prior to counting can influence the reported counts in this gear type.

Since the meso- and micro-zooplankton data overlap in sampled taxa, we investigated reported data biases of these two gear types by comparing taxa counted in both. We used EMP data, filtered to include only stations and dates in which both meso- and micro-zooplankton samples were collected. For each taxon (or life stage) represented in both datasets, we compared the total summed catch per unit effort (individuals per m$^3$ of water sampled; CPUE) from the mesozooplankton net (153 μm mesh, net) and the microzooplankton (43 μm mesh, pump) to assess where the data reported by each method may represent under-sampling. We did not compare the catch-efficiency of each gear type; our intent was to assess the reliability of data reported by each gear type to inform analyses of these data. In almost all cases, the two methods had drastically different total CPUEs, with the microzooplankton (pump) sample collecting substantially more individuals (19 out of 23 taxa; Fig 4). The mesozooplankton (net) sample was only better at capturing Cirripedia larvae, Cyclopoida adults, and *Oithona similis*

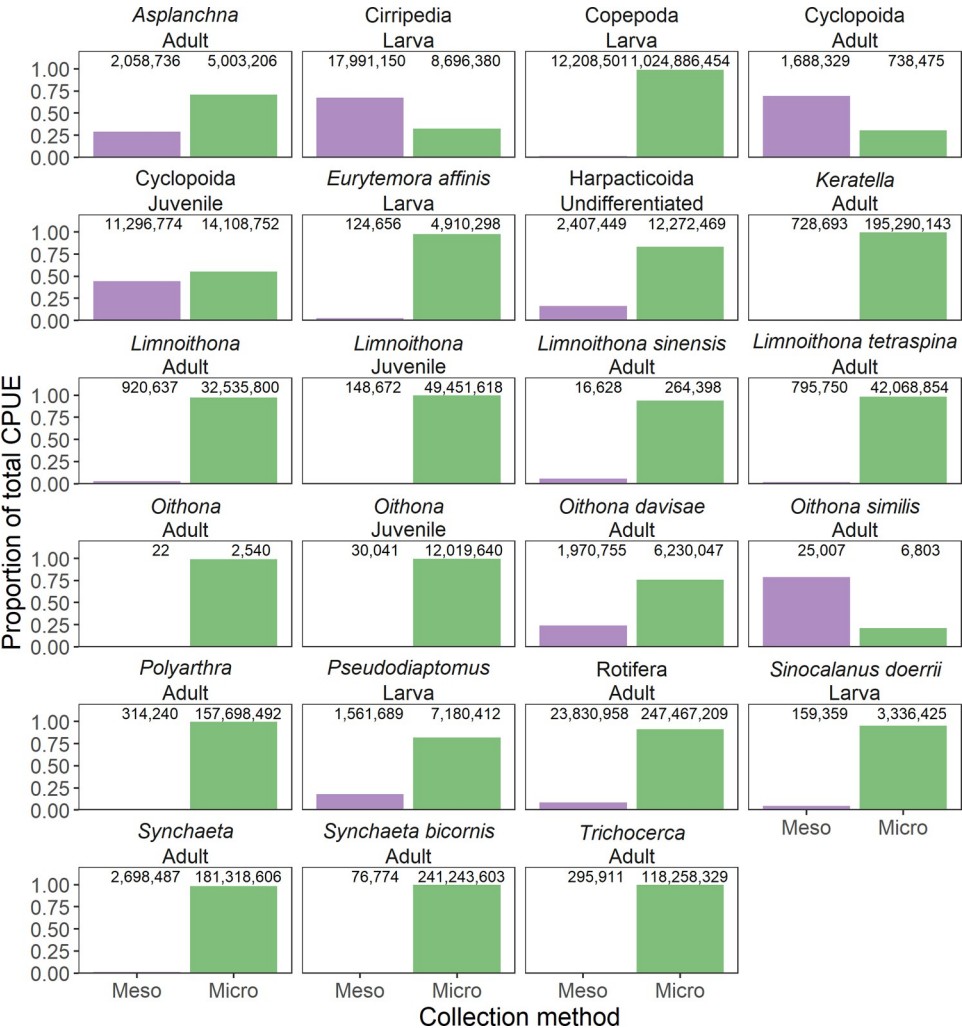

**Fig 4. Relative catch reported from two different collection methods.** The relative catch was calculated using data from the Environmental Monitoring Program (EMP): the 'Meso' (Mesozooplankton, 153 μm mesh, net) vs 'Micro' (Microzooplankton, 43 μm mesh, pump) sampling methods, for taxa counted in both. We did not compare the catch-efficiency of each gear type, our intent was to assess the reliability of data reported by each gear type to inform analyses of these data. Higher taxonomic levels (above species) represent "other" categories and do not include counts from the lower taxa they contain (e.g., *Synchaeta* Adult does not contain counts of the individuals from *Synchaeta bicornis* Adult). Numbers above bars indicate the total CPUE.

adults. The two catches were very similar for Cyclopoida juveniles (mesozooplankton/net captured 80% of the catch of microzooplankton/pump). Using this information, we developed a list of taxa and life stages under sampled by each method (excluding only Cyclopoida juveniles since the catch was so close), included as undersampled.csv. These under sampled plankton are retained in the integrated zooplankton dataset but can be flagged and removed using undersampled.csv. Flagged taxa should be analyzed with caution since their CPUEs are biased by an inappropriate gear type.

It is important to note that, prior to counting in the laboratory, the EMP microzooplankton (pump) samples are passed through a 154 μm sieve in laboratory and only the smaller size fraction is counted (under the assumption that the larger individuals are better sampled by the mesozooplankton/net sample). Thus, some of the under sampling of larger taxa by the microzooplankton (pump) samples may be an artifact of this laboratory methodology rather than an effect of the net mesh size. Therefore, these results may not apply to other zooplankton studies, but should be informative for users of this dataset.

**Field sampling techniques.** *Stepped-oblique net tow*. The most common sampling technique in these surveys is the stepped-oblique net tow (Table 2). In this method, the zooplankton net is attached to a metal sled. This sled may be solely used for meso- and macro-zooplankton (as in EMP and FMWT) or attached to a larger fish sampling net (as in 20mm and STN). The sled is deployed off the stern or side of a boat using a cable attached to a winch. The cable is spooled out to a standardized length based on the depth of the water. The boat proceeds at slow speeds while a specified amount of cable is slowly drawn in at specified time intervals following a tow schedule. As the cable is drawn in, the sled rises through the water in a stepwise fashion, sampling each strata of the water column approximately evenly.

*Horizontal net tow*. In a horizontal net tow, the net is held at a constant depth while the boat proceeds forward at slow speeds. FRP uses horizontal tows in which the net is held just below the surface of the water.

*Stationary sampling*. In some stations, FRP samples by holding the zooplankton net in a constant position and allowing the current to flow through the net for a pre-defined period of time [60]. This works best when sampling from shore or a stable structure, to attach the net to, and is most often used on ebb tides to sample water flowing out of a wetland.

*Pump*. Pumps are used by EMP for microzooplankton sampling [61]. Pumps are advantageous for microzooplankton because the filtered volume and net clogging are easier to monitor. However, larger organisms can escape the narrow mouth of a pump intake [62].

**Measurement of environmental variables.** *Salinity and temperature*. Specific conductivity and temperature are measured by all surveys using YSI probes (Yellow Spring Instruments, Inc, Yellow Springs, OH). Surface measurements are taken in the upper 100 cm of the water, while EMP, FMWT, STN, and 20mm also collect conductivity measurements at the bottom of the water column. For this dataset, we have converted conductivity to salinity using the ec2pss function from the wql package [63] for the R statistical programming language [64]. This function converts electrical conductivity to salinity using the Practical Salinity Scale 1978 for salinities between 2 and 42 [65] and the extension of the Practical Salinity Scale [66] for salinities below 2.

*Turbidity and Secchi depth*. All surveys measure Secchi depth, the depth at which a black and white disk is no longer visible from the surface. Secchi depth is recorded from the shaded side of the boat by an observer not wearing sunglasses. Secchi depth is inversely related to turbidity, which is measured by some surveys (20mm, FMWT, STN, and FRP) using YSI or Hach turbidity meters starting in more recent years.

**Target organisms identified.** Depending on the goals of the study, some surveys will enumerate different organisms than others, and identify them to a different level of taxonomic

resolution. For example, FMWT and EMP macrozooplankton samples are only processed for mysids and amphipods. Other invertebrates (insects, isopods, etc) are not counted. FRP macrozooplankton samples are processed for all macrozooplankton and micronekton, however insects are only identified to the family level, whereas mysids are identified to species. Comparing these three datasets requires understanding and accounting for these differences to avoid erroneously believing that FMWT or EMP samples had lower diversity than FRP samples (see *Data integration methods*). The list of taxa identified in each dataset is provided in taxa_lists.csv.

**Subsampling methods in the laboratory.** Due to the patchy distribution of zooplankton in the water column, most surveys collect relatively large samples and process a randomly selected subsample of the original sample. The accuracy of an abundance estimate based on a sample is directly related to the number of organisms counted, assuming they are randomly distributed with a Poisson distribution [62]. Therefore, the size of the original sample and proportion of the sample enumerated will determine accuracy of any derived abundance estimates. If one survey collects significantly larger samples or enumerates a higher number of individuals in its subsample, comparing abundance estimates between the two surveys could be confounded by their differing accuracies. In addition, differences in subsampling method can impact precision of an estimate [67]. For these surveys, subsampling is conducted with 1-ml pipetted aliquots for micro- and meso-zooplankton, and divider trays for macrozooplankton.

*Aliquots.* Mesozooplankton samples are typically subsampled with a micropipette (for specifics, see [61, 68]). The sample is first diluted to achieve a zooplankton concentration of between 200 and 400 organisms ml$^{-1}$. After mixing the sample in a beaker, a 1-ml subsample is drawn and pipetted onto a gridded Sedgewick-Rafter glass slide for identification under a microscope. Subsamples are processed until a target is reached, but these targets have changed over time. For EMP, FMWT, STN, and 20mm, the target was 200 total organisms from 1972 to 2003, 6% of the total dilution volume from 2004 to 2005, and from 2006 to present organisms were counted until 6% of the dilution volume had been processed and at least five and no more than 20 1-ml subsamples were processed [61, 68]. Under current methods and at the target concentration (200–400 organisms per ml), this results in at least 1,000 organisms and a maximum of 8,000 organisms counted per sample. When samples contain debris or detritus, dilution volume is often increased to enable staff to see all the organisms on a slide clearly, which results in lower total organism counts. FRP processes a minimum of five 1-ml subsamples until 400 organisms are counted, or 20 ml total, depending on which occurs first.

*Divider trays.* When samples are dense with organisms, macrozooplankton are also processed with subsampling using a sorting tray divided into quadrants (divider tray). targeting at least 220 organisms from 1972 to 1984 and at least 400 organisms from 1984 to present. In the divider-tray method, the sample is uniformly spread across a plastic tray and a 4-quadrant divider is then dropped on top of the tray. Taxonomists then enumerate only the invertebrates in the lower right-hand corner of the tray. For very heavy samples, this procedure may be repeated so that a 1/16th or a 1/64th fraction of the original sample is enumerated (for specifics, see [60, 61]). From 1972 to 1984, these surveys targeted a minimum count of 220 total organisms before subsampling is completed. From 1984 to present, 400 total organisms were targeted [60, 61], which gives a precision of +/- 10% [62].

**Calculations.** *Count per unit effort (CPUE).* CPUE calculations are based on the volume of water sampled. Most IEP surveys estimate volume using a flowmeter in the center of the net mouth (model 2030R, General Oceanics, Inc, Miami Florida). For EMP microzooplankton samples, the volume of the water pumped into the net is measured directly using a GPI inline digital flowmeter (Great Plains Industries, Inc, Sparta, NJ) near the output end of the hose

where water enters the net for filtration. The volume of water sampled is calculated by multiplying the flowmeter rotation count, a flowmeter constant relating number of counts to water speed, and the net mouth area. CPUE is reported in units of individuals m$^{-3}$.

*Biomass.* Meso- and microzooplankton biomass is most frequently calculated based on average weights derived from literature values. These calculations apply a single estimated value for mg carbon per individual for all individuals of a given life stage [69, 70]. There are no existing biomass values for many species, so related species must be used.

For mysids collected by EMP and FMWT, the first 100 individuals are also measured to the nearest mm. Biomass can then be calculated with length-weight regressions. Mysid length data and conversion equations are not currently included in the integrated dataset but should be included in future revisions.

We have compiled updated micro- and mesozooplankton biomass conversions from the literature into biomass_mesomicro.csv. All species and taxonomic groups are not covered, reflecting gaps in the literature, but these conversion values provide a starting point for researchers interested in estimating zooplankton biomass.

## Data integration methods

Data integration was completed in R version 4.0.3 [64]. All code is available in the R package zooper version 2.3.1 [56] and in the R script "Data_processing.R" within the data publication [55]. First, we created lookup tables to assist with the data integration. The locations of fixed sampling locations were compiled into stations.csv and the EMP moving entrapment zone (EZ) station locations were compiled into stations_EMP_EZ.csv. Taxonomic classifications were compiled into taxonomy.csv, while the taxonomic resolution of each source dataset and the dates this resolution had changed, or species were introduced, were compiled into taxa_lists.csv (Fig 3).

Datasets were downloaded from their sources online and reformatted for consistency by converting species codes to scientific names, renaming column names, converting units, and pivoting all datasets to the "long" format (where each row contains just one CPUE value for each taxon and sample). In some datasets, CPUE was reported as 0 in years before the taxa was counted at that taxonomic level. These values should have been recorded as missing values ("NA") because the abundances of those taxa were unknown before they were counted. To resolve this issue, in years when taxa were not counted, we replaced those incorrect 0s with "NA" values. However, non-native species were left with 0 CPUE before their known introduction year.

The consistent datasets were then bound together by column name. All environmental parameters were not measured by all datasets and those gaps are represented in the combined datasets with "NA" values. To reduce data duplication and file size, the combined dataset was then split into sample-level data (environment.csv; sampling location, date, environmental parameters, etc.) and zooplankton catch data (zooplankton.csv), each retaining the column "SampleID" as a key to rejoin them. The taxonomic resolution of each source dataset is unaltered and thus variable across surveys within the integrated dataset. Information on the taxonomic resolution of source datasets can be found in taxa_lists.csv (Fig 3).

**Resolving differences in taxonomic resolution: Methods used to create zooplankton_community.csv.** Differences in taxonomic resolution among studies could result in misleading findings from a community-level analysis of the integrated dataset. For example, EMP and FMWT lump all members of the genus *Tortanus* together and count them within the category *Tortanus* spp., while 20mm separates and counts *Tortanus discaudatus*, *Tortanus dextrilobatus*, and other *Tortanus* (*Tortanus* spp.). A naïve analysis would conclude that *Tortanus*

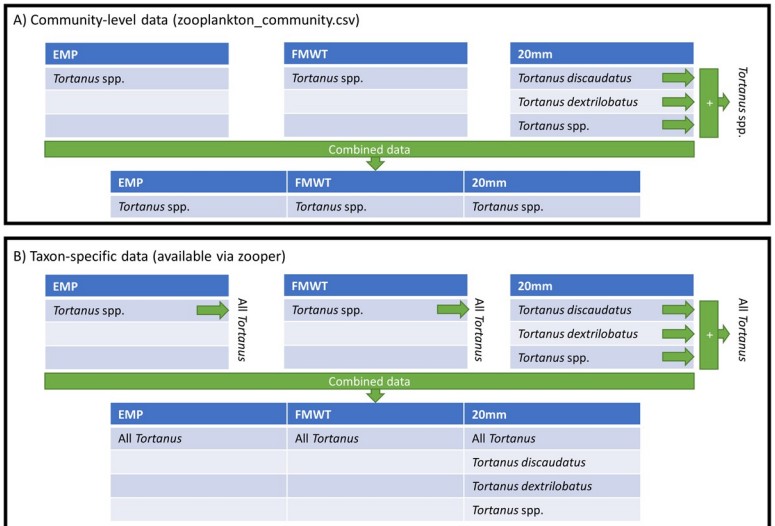

**Fig 5. Two methods used to resolve differences in taxonomic resolution among surveys.** Methods are illustrated using the genus *Tortanus* as an example, but issues with other taxa are resolved with the same methods. A) Taxonomic resolution is resolved to create a dataset for community-level analysis (available via zooplankton_community.csv or zooper). Since not all surveys measure *Tortanus* abundance to the species level, but all surveys measure *Tortanus* to the genus level or lower, the counts of *Tortanus* for all surveys are summed to the genus level (*Tortanus* spp.). The species-level data are then removed from the integrated dataset since their counts are included in *Tortanus* spp. B) Taxonomic resolution is resolved to provide all possible information on taxa of interest (available via zooper). All *Tortanus* counts are summed to an "All *Tortanus*" category, which is comparable across surveys. The individual *Tortanus* categories (*Tortanus* spp., *Tortanus discaudatus*, and *Tortanus dextrilobatus*) are retained in the integrated dataset.

*discaudatus* and *Tortanus dextrilobatus* only appear at the 20mm sampling locations while the lumped *Tortanus* spp. category is much more prevalent at EMP and FMWT sites. However, these results would be due solely to differences in taxonomic resolution among surveys. To resolve this issue, we developed a method to standardize taxonomic resolution to ensure data are comparable across surveys (Fig 5A). This method is applied to each individual zooplankton sample, so data from different samples are never aggregated together.

To start, we find all taxa that are not counted in every survey (*Tortanus discaudatus* and *Tortanus dextrilobatus* in the example above). Then, we sum these taxa up to a higher taxonomic level that is counted in all surveys (*Tortanus* spp.) and remove the lower taxa that have been summed (*Tortanus discaudatus* and *Tortanus dextrilobatus*) to prevent double counting. Now, all surveys have categories (*Tortanus* spp.) that represent the same set of taxa (all copepods in the genus *Tortanus*; Fig 5A). Any taxa that are not represented at a higher taxonomic level in all surveys are removed from the dataset. These removed taxa are less-commonly counted taxa such as Annelida, Nematoda, or Insecta. This process considers each life stage of a taxa separately and is applied separately to each size class, so taxa lists for microzooplankton samples are only compared to taxa lists for other microzooplankton samples (and the same for meso- and macrozooplankton). This solution has been applied to the zooplankton_community.csv table, which also has been merged with the environment.csv, taxonomy.csv, stations.csv, stations_EMP_EZ.csv, and undersampled.csv tables so that it is an analysis-ready dataset with all taxon- and sample-level information.

This method to resolve differences in taxonomic resolution will reduce the taxonomic resolution of more taxonomically-rigorous surveys. Unfortunately, this is unavoidable since taxonomic resolution can only be reduced to the least common denominator and cannot be increased over the resolution used in counting samples. If users wish to use the base taxonomic

resolution of specific surveys for a community-level analysis, they can select data from specific surveys from the taxonomically unaltered zooplankton.csv table.

**Other considerations and features of the zooper package.** In addition to differing taxonomic resolution among surveys, the taxonomic resolution has also changed over time within most surveys (all except FRP). In some cases, recently introduced taxa were added after their introduction, but in other cases taxa formerly identified to a higher level (e.g., genus) were subsequently identified to a lower level (e.g., species). Analyses of community change over time must take these changes in taxonomic resolution over time into account to prevent a naïve analysis from discovering increasing diversity over time that is solely attributed to changes in methods. The zooper R package [56] can correct for changes in taxonomic resolution over time by reducing the taxonomic resolution of the dataset to its lowest resolution at any point in time. However, this would exclude introduced species from analyses, so the package allows users to input a time-lag for introduced species. If surveys began counting an introduced species within a defined period of years (the time-lag) after its introduction, that species is retained in the time-corrected dataset.

In addition to the incorporation of a fix for changing taxonomic resolution over time, the zooper R package (and its associated interactive point-and-click shiny application) have a number of other options to customize your zooplankton dataset. These options allow users to filter the data by date, salinity, temperature, survey, size class, or sampling location. The taxonomic resolution fixes are then applied on the filtered dataset. This ensures the fewest possible alterations to the data are made. Lastly, the R package and shiny application also have an alternative solution for resolving differences in taxonomic resolution among studies. For users interested in querying all available data on certain taxa, the package will return all data on your chosen taxa along with summed categories representing higher taxonomic levels that are comparable across surveys (Fig 5B). Unlike the process used to create the zooplankton_community.csv file described above, this method does not remove lower taxonomic categories that are members of summed groups, so plankton are double-counted. Thus, outputs with this option selected should not be used for multivariate or community-level analyses. Users interested in using these advanced options to return a more customizable dataset are encouraged to produce their dataset with the zooper R package or shiny application, instead of using zooplankton_community.csv.

## Technical validation

The data from each component survey in the integrated dataset have undergone quality assurance and checking procedures. These procedures are similar in all five surveys and start in the field where each survey washes down the nets to ensure all organisms are captured, and then immediately preserves zooplankton samples after collection (in ethanol for FRP and in formalin for the other surveys).

Samples are processed in the laboratory by taxonomists who undergo a rigorous training and validation process. New taxonomists are trained to identify each taxon and their identifications of all organisms in individual subsamples (see *Subsampling methods in the* laboratory) are double-checked by a more senior taxonomist. As the number of mistakes committed by a new taxonomist decreases, so does the proportion of their subsamples verified by a senior taxonomist. At the highest level of qualification, 1/50 subsamples are checked for EMP, 1/20 subsamples are checked for FMWT/STN, 1/100 subsamples are checked for 20mm, and 1/5 subsamples are checked for FRP. FRP also sends 1/20 samples to an external laboratory for additional quality control.

After data are collected, each survey follows protocols to inspect the data for anomalous values. Flowmeter values used to calculate the volume sampled are of particular concern due to

their influence on the CPUE calculation. Flowmeter values are inspected for outliers and compared with values from other nets (zooplankton or fish) towed at the same time and nearby stations. If flowmeter values are deemed unreliable, they are replaced with an average from similar samples (same station or date). Dilutions and counts are inspected for outliers and the relationships among counts, volumes, and CPUE are inspected to further verify flowmeter and count values. Lastly, community composition and species abundances are verified against known distributions over space and time.

The data integration code within the R package zooper contains a number of tests to ensure the data processing and taxonomic solutions do not undermine the quality of the data. These tests ensure that each component function produces the expected output, no samples nor data rows are duplicated, and the same samples are present regardless of the taxonomic solution selected. We have also validated that the total zooplankton density is not incorrectly altered by the methods used to resolve differences in taxonomic resolution for community analyses (see *Resolving differences in taxonomic resolution: methods used to create zooplankton_community. csv*). To do this, we fit a linear regression to the total CPUE per sample from the taxonomically-fixed dataset, predicted by the same metric from the raw dataset. The total CPUE per sample from the taxonomically-fixed dataset was almost exactly identical to that in the unaltered source datasets (Fig 6, $R^2$ = 0.999999). The only differences were from taxa that were counted in some surveys but discarded in others, since these taxa are removed in the process to resolve the taxonomic resolution. When these problematic taxa were removed from both datasets, total CPUE per sample was exactly identical in all samples.

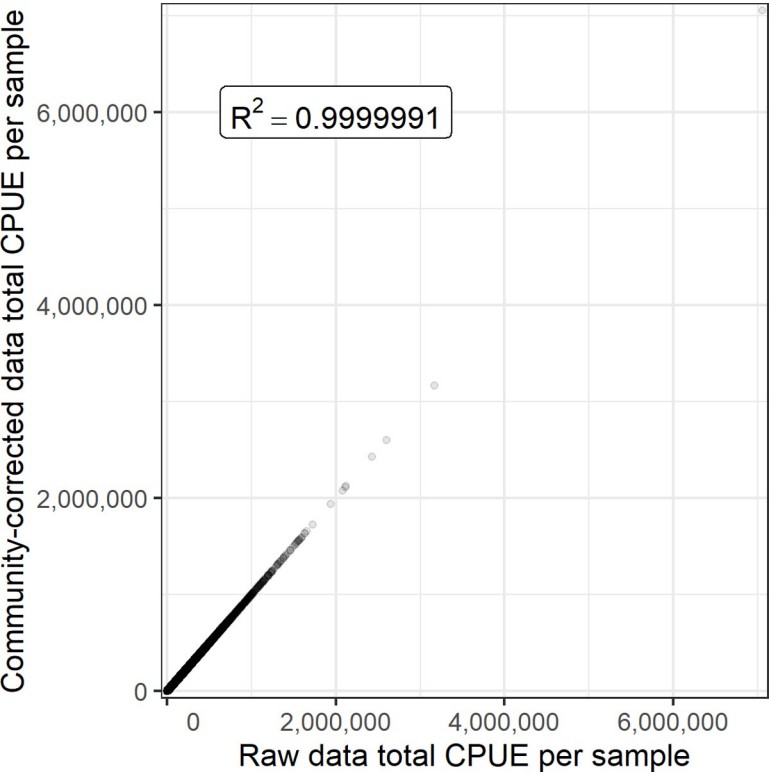

**Fig 6. Relationship between the raw CPUE in the source datasets and the CPUE after fixes to taxonomic resolution were applied (Fig 5A).** The total CPUE per sample was almost exactly identical in both cases, as indicated by the $R^2$ = 0.999999. Differences resulted from taxa that were counted in some surveys but discarded in others, since these taxa are removed in the process to resolve the taxonomic resolution. When these problematic taxa were removed from both datasets, total CPUE per sample was exactly identical in all samples.

This dataset will be updated as the component surveys publish new versions of their data. The zooper R package will be updated most frequently with each change in the underlying datasets, while new versions of the EDI data publication will be published approximately yearly when most component surveys have updated their published data.

## Discussion

### Comparison with existing datasets

Each of the component surveys collect other types of data not included in this integrated zooplankton dataset, but which may be useful in conjunction with the zooplankton data. FMWT, STN, 20mm, and FRP all collect fish abundance data, EMP and FRP collect abundance data on non-planktonic macroinvertebrates, and EMP collects many additional water quality parameters. Furthermore, many other long-term monitoring surveys collect water quality and fish abundance data in the SFE that may be useful for more complex modeling exercises in conjunction with this zooplankton dataset. Those surveys include the California Department of Fish and Wildlife Spring Kodiak Trawl [71]; the United States Fish and Wildlife Service Delta Juvenile Fish Monitoring Program [72] and Enhanced Delta Smelt Monitoring [73]; and the United States Geological Survey San Francisco Bay water-quality survey [74]. Related hydrological data (precipitation, flows, etc.) can be obtained from the Dayflow model of Delta Hydrology produced by the California Department of Water Resources (https://data.cnra.ca.gov/dataset/dayflow), the California Data Exchange Center (https://cdec.water.ca.gov/), or the United States Geological Survey National Water Information System [75].

Similar zooplankton community datasets from other estuaries are rare. The Scheldt estuary, Belgium has data available online from 1995–2018 (https://www.scheldemonitor.org/dataproducts/en/download/occurrence/dataset/1073), the Chesapeake Bay, USA has data available from 1984–2002 (https://www.chesapeakebay.net/what/downloads/baywide_cbp_plankton_database), and Kongsfjorden, Norway has data available from 1996–2016 (https://data.npolar.no/dataset/94b29b16-b03b-47d7-bfbc-1c3c4f7060d2). Zooplankton data from the Hudson River, New York, USA is available for 1987–2015 [15] but only at one location and with a coarse taxonomic resolution (cladocera adults, cladocera eggs, copepod adult, copepod nauplius, copepod egg, or rotifer). Even coarser data (grouped into mesozooplankton, macrozooplankton, zooplankton, and euphausiids) are available from the St. Lawrence Estuary in Canada from 2010–2019 [22]. Partial zooplankton community time series are available from Narragansett Bay, Rhode Island, USA (2001–2005, https://web.uri.edu/gso/research/plankton/data) and the North Inlet Estuary, South Carolina, USA (1981–1992, [16]). Other estuaries have monitored zooplankton, but data are not accessible online (see *Background and Motivation*).

### Data use and recommendations for reuse

While the full integrated dataset presented in this paper has not yet been used in any publications, the component datasets have been used in many studies of the SFE. Prior studies using the component datasets have investigated species invasions and their drivers [32, 33, 76–80], biotic and abiotic drivers of zooplankton abundance [38, 81–87], zooplankton population dynamics [88], and fish feeding and population dynamics [53, 89–95].

As with any dataset, users should familiarize themselves with the methods and limitations of the data before starting analyses. We attempted to distill the most important methodological details in this paper, but more details can be found along with the sources of each individual dataset [44–48]. Importantly, some methods have changed over time, which may impact the results of analyses. The spatiotemporal sampling effort of each survey has changed over time

due to changing management needs, unforeseen events, or shifting priorities. In addition, tax-
onomic resolution has increased in all studies except FRP (because it started so recently) as
more species were added to identification lists. In some cases, recently introduced taxa were
added after their introduction, but in other cases taxa formerly identified e.g., to the genus
level were subsequently identified to the species level. Analyses of community change over
time must take these changes in taxonomic resolution over time into account. The zooper R
package [56] and shiny app can correct for changes in taxonomic resolution over time (see
*Other considerations and features of the zooper package*).

Due to the long time series and history of environmental change, this dataset would be
well-suited to studies of climate change impacts, invasion ecology, water resources manage-
ment, anthropogenic nutrient or contaminant inputs, ecosystem declines, or wetland restora-
tion. Studies at the intersection of those topics would be particularly informative and fill key
knowledge gaps for management of this and other estuaries. For example, very few studies
have investigated the effects of climate change on zooplankton invasions [96] so this is a ripe
area of research that our long time series of a heavily invaded estuary [31] should be particu-
larly useful at addressing. Coupled with related datasets (see *Comparison with existing data-
sets*), our zooplankton dataset offers an opportunity to investigate trophic linkages and
cascades in a variable environmental context. The development of a food web model is a key
management need in the SFE [97]. The distribution of our data throughout the complex net-
work of waterways in the upper SFE, coupled with the complex water management and envi-
ronmental drivers determining flows, offers an interesting opportunity to investigate
metacommunity dynamics, dispersal, and ecohydrology. The relationship between flows and
zooplankton populations is another knowledge gap of active management interest in the SFE.
Lastly, the impact of climate change on zooplankton populations (either directly or indirectly
through forced changes to the managed hydrology) and how that would scale up to impact fish
populations would be an interesting and highly management-relevant avenue for research.
Taking advantage of these or other knowledge gaps, this comprehensive zooplankton commu-
nity dataset offers numerous opportunities for novel analyses to guide estuarine management
and further our understanding of ecological principles.

## Acknowledgments

Thanks to all the previous principal investigators of these studies and field and laboratory
workers from the California Department of Fish and Wildlife and the California Department
of Water Resources who collected and processed the samples for the data in this report. Special
thanks to Sally Skelton and Dorothy Crystal (retired CDFW) who processed 90% or more of
the samples included in this dataset. The component surveys were conducted by the Inter-
agency Ecological Program. We would also like to thank Steve Culberson and Louise Conrad
of the Delta Stewardship Council and the Interagency Ecological Program Science Manage-
ment Team for their support and encouragement in formalizing the technical team that made
this data integration project possible. Comments from Jesse Adams and Ted Flynn improved
an earlier version of this paper.

## Author Contributions

**Conceptualization:** Samuel M. Bashevkin, Rosemary Hartman, Karen Kayfetz.

**Data curation:** Samuel M. Bashevkin, Rosemary Hartman, Madison Thomas, Arthur Barros,
Christina E. Burdi, April Hennessy, Trishelle Tempel, Karen Kayfetz.

**Formal analysis:** Samuel M. Bashevkin.

**Methodology:** Samuel M. Bashevkin, Rosemary Hartman, Madison Thomas, Christina E. Burdi, Karen Kayfetz.

**Project administration:** Samuel M. Bashevkin.

**Software:** Samuel M. Bashevkin.

**Supervision:** Samuel M. Bashevkin.

**Validation:** Samuel M. Bashevkin.

**Visualization:** Samuel M. Bashevkin.

**Writing – original draft:** Samuel M. Bashevkin, Rosemary Hartman, Madison Thomas, Arthur Barros, Christina E. Burdi, April Hennessy, Trishelle Tempel, Karen Kayfetz.

**Writing – review & editing:** Samuel M. Bashevkin, Rosemary Hartman, Madison Thomas, Arthur Barros, Christina E. Burdi, April Hennessy, Trishelle Tempel, Karen Kayfetz.

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
