## [Decision Letter · Decision Letter 0]

11 Oct 2021

PONE-D-21-19296

Five decades (1972-2020) of zooplankton monitoring in the upper San Francisco Estuary

PLOS ONE

Dear Dr. Bashevkin,

Thank you for submitting your manuscript to PLOS ONE. After careful consideration, we feel that it has merit but does not fully meet PLOS ONE’s publication criteria as it currently stands. Therefore, we invite you to submit a revised version of the manuscript that addresses the points raised during the review process.

This manuscript has three reviewers. Among them, two reviewers suggested to reject and one reviewer suggested to accept. Please read all three reviewers' comments in detailed. Please revise this manuscript extremely careful. Particularly, the scientific contributions and many other issues should address in your revision point to point rebuttal letter. I will try to invite different reviewers to have second opinions.

We look forward to receiving your revised manuscript.

Kind regards,

Jiang-Shiou Hwang, Ph.D.

Academic Editor

PLOS ONE

Journal Requirements:

2. We note that Figure 1 in your submission contain [map/satellite] images which may be copyrighted. All PLOS content is published under the Creative Commons Attribution License (CC BY 4.0), which means that the manuscript, images, and Supporting Information files will be freely available online, and any third party is permitted to access, download, copy, distribute, and use these materials in any way, even commercially, with proper attribution. For these reasons, we cannot publish previously copyrighted maps or satellite images created using proprietary data, such as Google software (Google Maps, Street View, and Earth). For more information, see our copyright guidelines: http://journals.plos.org/plosone/s/licenses-and-copyright.

Additional Editor Comments (if provided):

This manuscript has three reviewers. Among them, two reviewers suggested to reject and one reviewer suggested to accept. Please read all three reviewers' comments in detailed. Please revise this manuscript extremely careful. Particularly, the scientific contributions and many other issues should address in your revision point to point rebuttal letter. I will try to invite different reviewers to have second opinions.

Reviewers' comments:

Reviewer's Responses to Questions

**Comments to the Author**

1. Is the manuscript technically sound, and do the data support the conclusions?

Reviewer #1: Partly

Reviewer #2: No

Reviewer #3: Partly

2. Has the statistical analysis been performed appropriately and rigorously? 

Reviewer #1: N/A

Reviewer #2: No

Reviewer #3: N/A

3. Have the authors made all data underlying the findings in their manuscript fully available?

Reviewer #1: Yes

Reviewer #2: No

Reviewer #3: Yes

4. Is the manuscript presented in an intelligible fashion and written in standard English?

Reviewer #1: Yes

Reviewer #2: No

Reviewer #3: Yes

5. Review Comments to the Author

Reviewer #1: The title reminded me of results on a long-term survey on SFE and some kinds of impacts of invasive non-indigenous organisms on the native ecosystem. However, in fact, this is a brief manual and explanation to use the software the authors created. Although separate datasets concerning long-term surveys on SFE are integrated in this, actual outputs should be shown by the authors as suggested in "Data Use and recommendation for reuse" in Discussion. So the authors had better indicate this in the website.

Reviewer #2: While the abstract engages in bragging about how this is the “longest available dataset (by 15 years) of estuarine zooplankton abundance worldwide,” except for 3 figures, the manuscript presents NO DATA on zooplankton patterns! It reads like a technical report to a funding agency rather than a scientific paper. Further, the manuscript is not well-written, particularly for authors who are presumably writing in their first language. Also, the statement that “long-term zooplankton datasets from estuaries are less common” ignores several unmentioned estuarine zooplankton datasets in published papers that should have been cited (see recent issues of Journal of Plankton Research and papers cited therein). In short, this manuscript appears to have been written by authors who are inexperienced in writing scientific papers. While producing the long-term dataset is certainly commendable, its presentation deserves better than this manuscript, which is a disservice to the authors, and their colleagues and predecessors who devoted considerable effort to collecting and analyzing the data.

In terms of writing, there is inconsistency in the hyphenation of double-word adjectives, and the same words, or combinations of words, are frequently presented differently, and often incorrectly (“under sampled” versus “undersampled”, “lifecycle”, “timeseries”). In the tables, abbreviations for surveys should be given in the table legend of each table, not just once in the text. Pronouns that begin sentences (such as “They”) should refer to the last noun used in the previous sentence, not one used before the last noun. Slang such as “lab” instead of “laboratory” should be avoided. Descriptions of sampling methodology should be written in past tense (although the project appears to be ongoing, the data for this manuscript ended in 2020).

The length of the introductory material should be greatly reduced. The Introduction begins on p. 2, with “Background and Motivation” (p. 2-7), “Monitoring overview” (p. 7-10, “Dataset description and access” (p. 10-14), followed by “Methods” (with various subtitles) from p. 14-31. THERE IS NO “RESULTS” SECTION. The manuscript then concludes with sections for “Discussion” (p. 32-34), “References (p. 34-42), and several figures.

Reviewer #3: good and important work to put the different data sets together. In the abstract, line 30, "what make the fishes important?" (commercially important?). Then the UK people have similar data sets from around their islands and inland lakes. And Hawaii has the Hawaii Ocean Time-series (HOT) from their station ALOHA. One day one can find correlations like:

North–South shifts of the Gulf Stream and their climatic connection with the abundance of zooplankton in the UK and its surrounding seas --- Arnold H. Taylor

ICES Journal of Marine Science, Volume 52, Issue 3-4, June 1995, Pages 711–721, https://doi.org/10.1016/1054-3139(95)80084-0

6. PLOS authors have the option to publish the peer review history of their article (what does this mean?). If published, this will include your full peer review and any attached files.

Reviewer #1: No

Reviewer #2: No

Reviewer #3: No

---

## [Author Response · Author response to Decision Letter 0]

12 Jan 2022

See attached response to reviewers document

---

## [Decision Letter · Decision Letter 1]

26 Jan 2022

PONE-D-21-19296R1

Five decades (1972-2020) of zooplankton monitoring in the upper San Francisco Estuary

PLOS ONE

Dear Dr. Bashevkin,

Thank you for submitting your manuscript to PLOS ONE. After careful consideration, we have decided that your manuscript does not meet our criteria for publication and must therefore be rejected

I am sorry that we cannot be more positive on this occasion, but hope that you appreciate the reasons for this decision.

Yours sincerely,

Jiang-Shiou Hwang, Ph.D.

Academic Editor

PLOS ONE

Reviewers' comments:

Reviewer's Responses to Questions

**Comments to the Author**

1. If the authors have adequately addressed your comments raised in a previous round of review and you feel that this manuscript is now acceptable for publication, you may indicate that here to bypass the “Comments to the Author” section, enter your conflict of interest statement in the “Confidential to Editor” section, and submit your "Accept" recommendation.

Reviewer #1: All comments have been addressed

2. Is the manuscript technically sound, and do the data support the conclusions?

Reviewer #1: No

3. Has the statistical analysis been performed appropriately and rigorously? 

Reviewer #1: Yes

4. Have the authors made all data underlying the findings in their manuscript fully available?

Reviewer #1: Yes

5. Is the manuscript presented in an intelligible fashion and written in standard English?

Reviewer #1: Yes

6. Review Comments to the Author

Reviewer #1: Although I understand the value of the paper, I wonder if it is published in an international journal. This paper totally depends on some independent long-term projects. So, as previously mentioned, this is actually a manual to comprehensively deal with previous data. As you suggested at L705-715, could you concretely indicate some kinds of perspectives using this data set?

7. PLOS authors have the option to publish the peer review history of their article (what does this mean?). If published, this will include your full peer review and any attached files.

Reviewer #1: No

- - - - -

---

## [Author Response · Author response to Decision Letter 1]

3 Feb 2022

See attached responses to reviewers, but note a reviewer has already reviewed this draft and stated "All comments have been addressed."

---

## [Decision Letter · Decision Letter 2]

22 Feb 2022

PONE-D-21-19296R2Five decades (1972-2020) of zooplankton monitoring in the upper San Francisco EstuaryPLOS ONE

Dear Corresponding author, Thank you for submitting your manuscript to PLOS ONE. After careful consideration, we feel that it has merit but does not fully meet PLOS ONE’s publication criteria as it currently stands. Therefore, we invite you to submit a revised version of the manuscript that addresses the points raised during the review process. Pls. also make use in following an amended copy of your submission sent by REVIEWER 4!

We look forward to receiving your revised manuscript.

Kind regards,

Hans-Uwe Dahms, Ph.D.

Academic Editor

PLOS ONE

Journal Requirements:

Additional Editor Comments:

According to REVIEWER 4, pls.:

1. Figure 1. increase the x axis and y axis font size, the sampling points can be more well defined.

2. Include a line or two about the knowledge gap and future recommendations.

3. Rainfall plays huge role in the estuarine ecosystem, why was there no data about rainfall?

Reviewers' comments:

Reviewer's Responses to Questions

**Comments to the Author**

1. If the authors have adequately addressed your comments raised in a previous round of review and you feel that this manuscript is now acceptable for publication, you may indicate that here to bypass the “Comments to the Author” section, enter your conflict of interest statement in the “Confidential to Editor” section, and submit your "Accept" recommendation.

Reviewer #4: (No Response)

Reviewer #5: All comments have been addressed

2. Is the manuscript technically sound, and do the data support the conclusions?

Reviewer #4: Yes

Reviewer #5: Yes

3. Has the statistical analysis been performed appropriately and rigorously? 

Reviewer #4: Yes

Reviewer #5: Yes

4. Have the authors made all data underlying the findings in their manuscript fully available?

Reviewer #4: Yes

Reviewer #5: Yes

5. Is the manuscript presented in an intelligible fashion and written in standard English?

Reviewer #4: Yes

Reviewer #5: Yes

6. Review Comments to the Author

Reviewer #4: PONE-D-21-19296R2 - Five decades (1972-2020) of zooplankton monitoring in the upper San Francisco Estuary

This manuscript presents the dataset of estuarine zooplankton abundance worldwide from 1972-2020. This manuscript is very informative, the background is well written to set the premises of the dataset. The methodology is well explained. I also appreciate the inclusion of micro-zooplankton; these organisms are ecologically important but are often understudied. This dataset will be handy for ecological synthesis and ecosystem modellers. I recommend this manuscript for publication. However, I have some suggestions and queries

1. Figure 1. increase the x axis and y axis font size, the sampling points can be more well defined.

2. Include a line or two about the knowledge gap and future recommendations.

3. Rainfall plays huge role in the estuarine ecosystem, why was there no data about rainfall?

Reviewer #5: Although, one of the reviewers talked about correlating the dataset with climate conditions, which can be done as another work. but I was at least expecting to see an interpretation of the result/data presented, or what the data show.

7. PLOS authors have the option to publish the peer review history of their article (what does this mean?). If published, this will include your full peer review and any attached files.

Reviewer #4: No

Reviewer #5: **Yes: **Esther U. Kadiene

---

## [Author Response · Author response to Decision Letter 2]

24 Feb 2022

Please see attached "response to reviewers" document.

---

## [Editor Report · Decision Letter 3]

2 Mar 2022

Five decades (1972-2020) of zooplankton monitoring in the upper San Francisco Estuary

PONE-D-21-19296R3

Dear Dr. BASHEVKIN,

We’re pleased to inform you that your manuscript has been judged scientifically suitable for publication and will be formally accepted for publication once it meets all outstanding technical requirements.

Kind regards,

Hans-Uwe Dahms, Ph.D.

Academic Editor

PLOS ONE

Additional Editor Comments (optional):

This DATA SET contribution has now reached an ACCEPTABLE level.

---

## [Editor Report · Acceptance letter]

4 Mar 2022

PONE-D-21-19296R3 

Five decades (1972-2020) of zooplankton monitoring in the upper San Francisco Estuary 

Dear Dr. Bashevkin:

I'm pleased to inform you that your manuscript has been deemed suitable for publication in PLOS ONE. Congratulations! Your manuscript is now with our production department. 

Kind regards, 

on behalf of

Dr. Hans-Uwe Dahms 

Academic Editor

PLOS ONE